# Macroeconomic Shocks and Changing Dynamics of the U.S. REITs Sector

**Rangan Gupta [1] , Zhihui Lv [2],\* and Wing-Keung Wong [3,4,5]**

[1]  Department of Economics, University of Pretoria, Pretoria 0002, South Africa; rangan.gupta@up.ac.za
[2]  KLASMOE & School of Mathematics and Statistics, Northeast Normal University, Changchun 130024, China
[3]  Department of Finance, Fintech Center, and Big Data Research Center, Asia University, Taichung 41354, Taiwan; wong@asia.edu.tw
[4]  Department of Medical Research, China Medical University Hospital, Taichung 40402, Taiwan
[5]  Department of Economics and Finance, The Hang Seng University of Hong Kong, Shatin 999077, Hong Kong, China
\*  Correspondence: luzh694@nenu.edu.cn

**Abstract:** Unlike the existing literature, which primarily studies the impact of only monetary policy shocks on real estate investment trusts (REITs), this paper develops a change-point vector autoregressive (VAR) model and then analyzes, for the first time, regime-specific impact of demand, supply, monetary policy, and spread yield shocks, identified using sign-restrictions, on US REITs returns. The model first isolates four major macroeconomic regimes in the US since the 1970s and discloses important changes to the statistical properties of REITs returns and its responses to the identified shocks. A variance decomposition analysis revealed aggregate supply shocks to have dominated in the early part of the sample period, and monetary policy and spread shocks at the end. Our results imply that ignoring other possible shocks in the model is likely to lead to incorrect inferences, and over-reliance on (conventional) monetary policy in correcting for possible bubbles in the REITs sector, which it will fail to rectify, given the importance of other shocks driving the REITs sector.

**Keywords:** change-point VAR model; macroeconomic shocks; US REITs sector

## 1. Introduction

The rapid decline in real estate prices, following a prolonged boom, is commonly associated as the main underlying reason for the global economic and financial crisis of 2008 to 2009 [1,2]. Naturally, understanding what macroeconomic and financial shocks drives the real estate market is of paramount importance, especially for a policymaker aiming to avoid future catastrophic effects observed under the 'Great Recession'. While there exists a large number of studies that have analyzed the role of both conventional and unconventional (in the wake of the zero lower bound (ZLB) scenario) monetary policy [3–8], and more recently, fiscal policy shocks on real estate markets [9,10], and the feedback from it in shaping policy decisions, there is dearth of studies that have analyzed the role of aggregate demand, aggregate supply, and bond yield spread shocks on real estate markets. The only exception to this is the recent paper by Plakandaras et al. [11], where the authors study the effect of macroeconomic shocks in the determination of house prices in the US and the UK by employing time-varying parameter vector autoregressive (TVP-VAR) models covering the historical annual periods of 1830 to 2016 and 1845 to 2016, respectively. From the examination of the impulse responses of house prices on macroeconomic shocks, Plakandaras et al. [11], found that technology shocks dominate in the U.S. real estate market, while their effect is unimportant in the U.K, where, monetary policy shocks drives most of the house price evolution.

Against this backdrop, realizing that housing markets are regional in nature, with tremendous heterogeneity in terms of their response to (monetary) policy shocks [12–14], we analyze the role of various macroeconomic shocks in driving the Real Estate Investment Trusts (REITs) prices of the US, which tends to be homogenous across the country, being based on a broad single index. For our purpose, unlike Plakandaras et al. [11], we use higher-frequency (monthly) data covering the more recent period 1972:12 of 2016:12. The use of monthly data allows us to identify the shocks in a relatively cleaner manner [15,16], based on a change-point vector autoregressive (VAR) model that allows for different regimes throughout the sample period and identifies a variety of shocks (supply, demand, monetary policy, and the spread between long- and short-run maturities), derived from the theoretical reactions of an innovative general equilibrium model developed by Liu et al. [17]. This approach enables the VAR model to endogenously identify changes to the structure of the real estate market, as widely discussed in Simo-Kengne et al. [18], as well as variations to the properties of exogenous shocks during the sample period. We prefer to use this approach over the complete TVP-VAR framework used by Plakandaras et al. [11], as we want to explicitly identify regimes over our high-frequency data, rather than assuming that each point in time is a separate regime, as done in TVP-VARs—which is, perhaps, more appropriate at lower frequency data (like quarterly and annual). It is important to note that the spread shock is important for us, since the time period of our analysis involves the period of ZLB and hence, that of unconventional monetary policy, which in turn involved compression in the long-term yield spread (Beginning in the summer of 2007, money markets around the world experienced sustained periods of dysfunction with sharply higher short-term interest rates for commercial paper and interbank borrowing. This intense liquidity squeeze led the Federal Reserve (Fed) to substantially lower its Federal funds rate (FFR) and act as the liquidity provider of last resort to supply funds to banks and the broader financial system via its Term Auction Facility (TAF). The FFR, the Fed's traditional policy instrument, reached its effective ZLB in December 2008, and the Fed faced the challenge of how to further ease the stance of monetary policy as the economic outlook deteriorated. While the FFR had reached its effective ZLB, large-scale asset purchases (LSAPs), which reduced the supply of riskier long term assets and increased the supply of safer liquid assets (bank reserve), causing the spread to decline.). To the best of our knowledge, this is the first attempt to analyze the role of various macroeconomic and financial shocks over and above the monetary policy shock, in driving the REITs prices based on a change-point VAR. In this regard, it must be mentioned that the role of macroeconomic news surprises, along with monetary policy surprises for real estate markets have been recently studied by Marfatia et al. [19] and Nyakabawo et al. [20], based on single-equation approaches. However, these studies do not make an attempt to identify these shocks in a structural fashion based on a VAR model, and hence, cannot track the impact of these shocks over time, but just its correlation with the real estate returns (and volatility).

To summarize, given the importance of the real estate sector in the recent financial crisis, it is important to study the role of macroeconomic shocks driving the sector. However, existing studies have primarily concentrated on the role of monetary policy shocks and, in this regard, we deviate from the current literature by developing a change-point VAR model and then analyzing, for the first time, regime-specific impact of demand, supply, monetary policy, and spread yield shocks, identified using sign-restrictions, on US REITs returns. As indicated, based on our analysis over the monthly period of 1972:12 to 2016:12, aggregate supply shocks have dominated in the early part of the sample period, and monetary policy and spread shocks at the end. Our results imply that ignoring other possible shocks in the model is likely to lead to incorrect inferences, and over-reliance on (conventional) monetary policy in correcting for possible bubbles in the REITs sector, which it will fail to rectify, given the importance of other shocks driving the REITs sector. The remainder of the paper is organized as follows: Section 2 presents the data and the methodology, with Section 3 discussing the results, and Section 4 concluding the paper.

## 2. Data and Methodology

### 2.1. Sign Restrictions and Data

In this model, according to Kapetanios et al. [21] and Liu et al. [17], we investigated the four structural shocks: The monetary policy shock, the spread shock, the demand shock, and the supply shock and six variables: The short-term nominal interest rate ($I_t$), the REITs returns ($R_t$), the unemployment rate ($U_t$), the money holdings ($M_t$), the price inflation ($\pi_t$), and the interest rate spread $S_t$. We imposed the sign restrictions on the first period reaction of the VAR model and set the nominal interest rate such that it did not react to shocks during the financials crisis. The sign restrictions are presented in Table 1.

**Table 1.** Sign restrictions.

| Shock\Variables | $I_t$ | $R_t$ | $U_t$ | $M_t$ | $\pi_t$ | $S_t$ |
|---|---|---|---|---|---|---|
| Monetary policy shock | ≥ | ≤ | ≥ | ≤ | ≤ | ≤ |
| Spread shock | ≤ | ≥ | ≥ | ≥ | ≤ | ≤ |
| Demand shock | ≥ | ≥ | ≤ | ≤ | ≥ | ≥ |
| Supply shock | ≥ | ≥ | ≤ | ? | ≤ | ≤ |

Note: The sign '≥' denotes a positive response, the sign '≤' denotes a negative response, and '?' denotes an uncertain response that the sign can either be positive or negative, decided by the calibration of the model.

To derive the sign restrictions to impose on the change-point VAR model, we used the results derived from the New Keynesian Dynamic Stochastic General Equilibrium (NKDSGE) model of Liu et al. [17] to determine how each variable reacts to shocks. In the aftermath of a positive monetary policy shock, the real money holdings fall due to the higher cost of holding money. The increase in the nominal interest rate leads to a rise in the demand for short-term bonds that generates a fall in real activity and REITs prices and a rise in unemployment. The fall in output generates an increase in the long-run nominal interest rate, which is lower than the increase in the short-run interest rate, and hence the spread between long- and short-run interest rates falls. A positive interest rate spread shock increases the long-run interest rate, which generates a fall in consumption and output, as implied by the demand for long-term bonds. The contraction in output induces REITs prices to fall and unemployment to rise. The fall in output generates a reduction in inflation, as implied by the Phillips curve, whose effect is to induce a fall in the short-run interest rate (through the Taylor rule), which causes real money holdings to increase since the money demand is negatively related to the short-run nominal interest rate. The direct effect of the demand shock is to increase real activity and REITs prices with unemployment falling, since this shock actually increases the marginal utility of consumption for any given level of consumption, which in turn also increases the long-term nominal interest rate sharply due to the demand for long-term bond. The rise in output growth generates an increase in inflation, whose effect is to increase the short-term nominal interest rate, causing real money holdings to fall. The spread increases, since the increase in the long-term interest rate is stronger than the corresponding short-term interest rate increase. Finally, the aggregate supply shock produces an increase (decrease) in output (inflation), which results in increases in REITs prices and a decrease in unemployment. Output growth rises sharply and, therefore, induces the central bank to raise the short-term nominal interest rate, which generates a fall in the interest rate spread, and possibly also real money balances, with the final effect on the latter uncertain due to the positive income effect.

We collected data for the effective FFR, 10-Year Treasury bond yield at constant maturity, civilian unemployment rate, consumer price index (CPI), M2 definition of the money supply, and the monthly total returns index for the FTSE Nareit U.S. ALL REITs. The ALL REITs index is a market capitalization-weighted index that includes all tax-qualified REITs listed on the New York Stock Exchange, the American Stock Exchange, or the NASDAQ National Market List. We took data from the FRED database of the Federal Reserve Bank of St. Louis for all variables, barring the REITs index, which, in turn, was obtained from www.reit.com. Our sample covers the monthly period from 1972:12 to

2016:12, with the starting date driven by data availability at the time of writing this paper. We note that the end date is in line with one year from the end of the unconventional monetary policy regime, given that it is widely-believed that monetary policy takes over a year to affect the macroeconomic variables. In addition, the end date is also aligned with tapering of policies aimed towards stimulating the real estate market directly. The unemployment rate, CPI, and M2 were seasonally adjusted. The interest rate spread is defined as the 10-year yield minus the FFR. We used the 12-month percentage change to compute inflation, the growth rate of M2, and the growth rate of REITs prices.

### 2.2. Change-Point VAR Model

The model used in this paper is based on the change-point VAR model developed by Chib [22,23]. The detailed description is as follows:

$$Z_t = C_s + \sum_{j=1}^{K} B_s Z_{t-j} + \theta_s^{\frac{1}{2}} \varepsilon_t \tag{1}$$

where $B_s$ and $\theta_s$ are the autoregressive coefficients depended on the regime and reduced form variance covariance matrices, the state variable $s$ is modeled as a discrete time and follow an $M$ state Markov chain with the transition probability matrix, and the transition probability matrix specifies s to be either stay at the current value or switch to the next higher value, but not allow to switch back to past regimes. The restricted transition probabilities matrix $P$ is a diagonal matrix with the diagonal elements

$$p_{ij} = p(S_t = j | S_{t-1} = i) \tag{2}$$

where

$$p_{ij} > 0 \ if \ j = i + 1 \ or \ i = j$$

$$p_{MM} = 1$$

$$p_{ij} = 0 \ otherwise.$$

And in this paper, according to Liu et al. [17], it is suitable for us to set $M = 3$. The readers can refer to Chib [22,23] and Liu et al. [17] for more details. Thus, the restricted transition probability matrix is denoted by

$$\widetilde{P} = \begin{pmatrix} p_{11} & 0 & 0 & 0 \\ 1 - p_{11} & p_{22} & 0 & 0 \\ 0 & 1 - p_{22} & p_{33} & 0 \\ 0 & 0 & 1 - p_{33} & 1 \end{pmatrix} \tag{3}$$

### 2.3. Setting Dummy Observations

Based on the literature [17,24–26], we constructed the following matrices of dummy observations:

$$Y_D = \begin{pmatrix} \frac{diag(\gamma_1 \sigma_1 \cdots \gamma_N \sigma_N)}{\tau} \\ 0_{N \times (P-1) \times N} \\ \cdots \\ diag(\sigma_1 \cdots \sigma_N) \\ \cdots \\ 0_{1 \times N} \end{pmatrix} \text{ and } X_D = \begin{pmatrix} \frac{J_P \otimes diag(\gamma_1 \sigma_1 \cdots \gamma_N \sigma_N)}{\tau} & 0_{NP \times 1} \\ 0_{N \times NP} & 0_{N \times 1} \\ \cdots & \cdots \\ 0_{1 \times NP} & c \end{pmatrix} \tag{4}$$

where the matrix $J_P$ stands for $diag(1, 2, \cdots, P)$, $\tau = 0.03$, and $c = 1$ standing for the tightness of the prior on the VAR coefficients and the constant terms, respectively [25]. We obtain the estimates of both $\sigma_i$ and $\gamma_i$ from the AR(1) model via Ordinary Least Squares (OLS) for $i = 1, 2, \cdots, N$, where $\sigma_i$ stands for the standard deviation of the residual and $\gamma_i$ is the estimate of the AR(1) coefficient.

## 2.4. Priors for VARs

We assume the priors of the coefficients to follow a normal distribution and the covariances to follow an inverted Wishart distribution, with the first $M$ regimes assumed to follow a Normal Inverse Wishart prior [27,28]. The traditional approach is to assume that the priors of the coefficients follow the Multivariate Gaussian distribution with the vector zero mean and the covariance matrix with $diag(1000, 1000, \cdots, 1000)$. Howevwe, in this paper, to be consistent with the unconventional monetary policy regime and to ensure the lagged coefficients on the non-dependent variables in the interest rate equation are close to zero, we imposed the prior of the coefficients in this period to follow Multivariate Gaussian distribution with zero mean. In addition, the covariance matrix with the diagonal elements corresponding to the coefficients of interest in the interest rate equation was set to the value of $1 \times 10^{-12}$, and the remaining diagonal elements were set to be 1000, and other elements were set to zero.

## 2.5. Priors for the Transition Probability

We assume the priors for the nonzero elements of the transition matrix $p_{ij}$ follow the Dirichlet distribution, namely $p_{ij}^0 = D(u_{ij})$, where $D(\cdot)$ denotes the Dirichlet distribution and

$$u_{ij} = \left\{ \begin{array}{l} 10 \ i = j \\ 1 \ i \neq j \end{array} \right. \tag{5}$$

In this model setting, one could easily show that the posterior distribution follows the Dirichlet distribution:

$$p_{ij} = D(u_{ij} + \pi_{ij}) \tag{6}$$

where $\pi_{ij}$ denotes the times of regime $i$ that is followed by regime $j$.

In addition, we tool the sample by using the Gibbs sampling algorithm, estimate the change-point VAR model with the 200,000 replications, and deleted the first 195,000 replications so that the remaining replications became more random. Readers can refer to Chib [22,23], Kim and Nelson [29] and Liu et al. [17] for more details.

## 3. Empirical Results

Figure 1 shows the estimated probability of the four regimes, with them corresponding to the periods of June 1973 to January 1985, February 1985 to January 2009, February 2009 to May 2013, and June 2013 to December 2016. The VAR model was estimated based on six lags, as suggested by the Bayesian Information Criterion (BIC), and hence the first regime starts from June, 1972. The estimate for the first breakpoint is consistent with financial liberalization in the US, with the second one corresponding to the end of the global financial crisis, and the third one with the 'tapering' of the unconventional monetary policies.

To tie these breakpoints to changing macroeconomic dynamics, Figure 2 plots some key reduced form summary statistics from the change-point VAR. The top panel of Figure 2 presents the persistence of each of the endogenous variables in the change-point VAR for each regime. The FFR, bond yield spread, and M2 growth performed very similarly throughout the sample period. These variables increased from regime 1 to regime 2 and decreased during regime 3, and then increased in regime 4. Moreover, the unemployment rate showed a similar trend in the first three regimes. The persistence of inflation and REITs return changed over all four regimes, and during the financial crisis, the persistence of inflation reached its lowest value, while the persistence of REITs returns reached its highest value.

The second panel of Figure 2 shows the diagonal elements of the error covariance matrix in each regime. We observed that the volatility of the reduced-form errors declined for all the time series, with a different extent from regime 1 to regime 2. Furthermore, during the last two regimes covering the financial crisis and unconventional monetary policies due to the ZLB, the volatility of the reduced-form errors increased for all the variables, except for the bond yield spread and the FFR.

The last panel of Figure 2 shows the unconditional volatility of each variable in each regime. The volatility of the bond yield spread, and CPI inflation were extremely similar throughout the whole sample period. In general, all the plots were consistent to the second panel, except the behavior of the CPI inflation in regime 3.

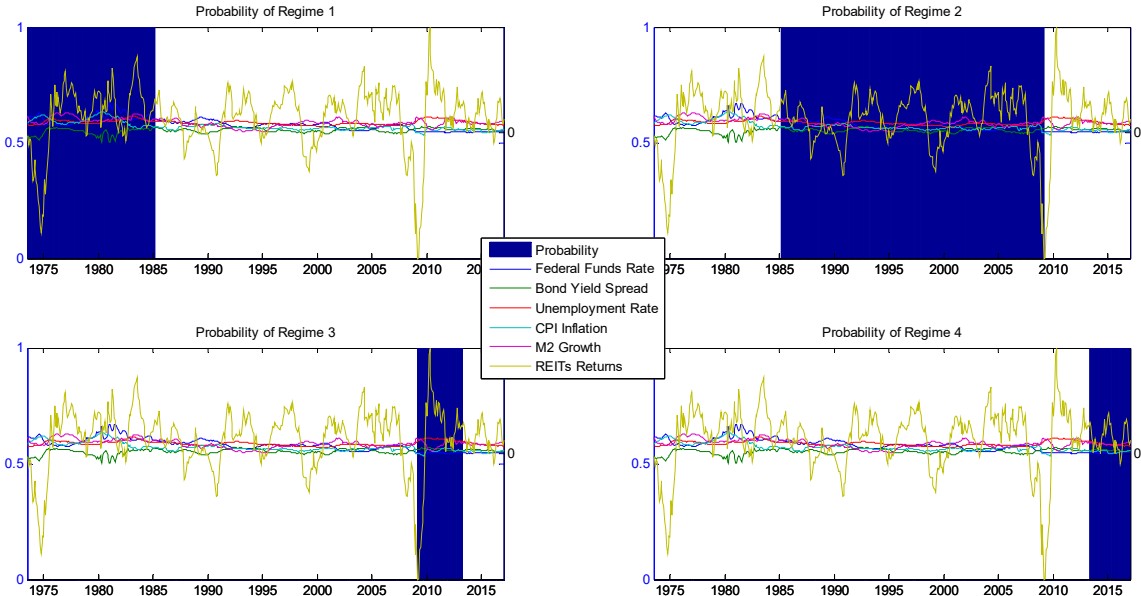

**Figure 1.** The estimated probability of each regime. Notes: The four regimes correspond to the periods June 1973 to January 1985, February 1985 to January 2009, February 2009 to May 2013, and June 2013 to December 2016.

The empirical framework is well-suited to investigate changes in macroeconomic dynamics across the sample horizon, since the change-point VAR model allows the coefficients in the model to vary across regimes. Figures 3–6 plot the impulse response functions (IRFs) of the six endogenous variables to a one-standard-deviation shock for the four identified shocks across the four regimes. We obtained the median and 68% confidence bands based on 5000 retained Gibbs replications. Since our focus is the REITs returns, we concentrate on discussing the effect of the various shocks on this variable. The effects on the other variables for the four shocks were similar to those obtained by Liu et al. [17], and the reader can refer to that paper for a more detailed discussion.

Figure 3 presents the responses of the variables to a contractionary monetary policy shock (i.e., an increase in the nominal interest rate). For the third regime, this shock was absent since the nominal interest rate was set at approximately zero, corresponding to the ZLB scenario. The size of the negative impact was quite similar in regimes 1 and 2, though it was relatively stronger under regime 1, with the effect being significant for about a year. The recovery was faster in regime 1 relative to regime 2, though the effect in the former regime increased again from three-years onwards. However, the strongest negative effect was observed in regime 4, with the effect being significant for over one and half years. This strong impact is probably an indication of the recovery that took place in the US real estate sector, and the economy in general, post the 'Great Recession'. In all cases, however, the effect on REITs returns was negative over the entire horizons of the 50 months considered.

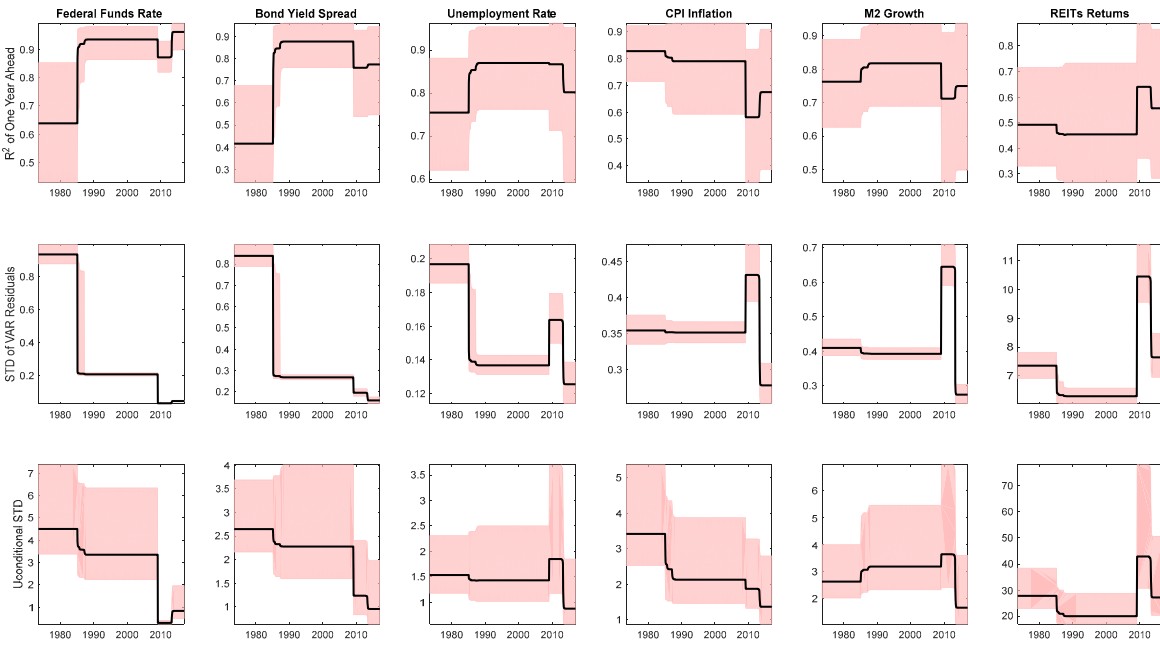

**Figure 2.** Regime dependent summary statistics; Note: See Notes to Figure 1.

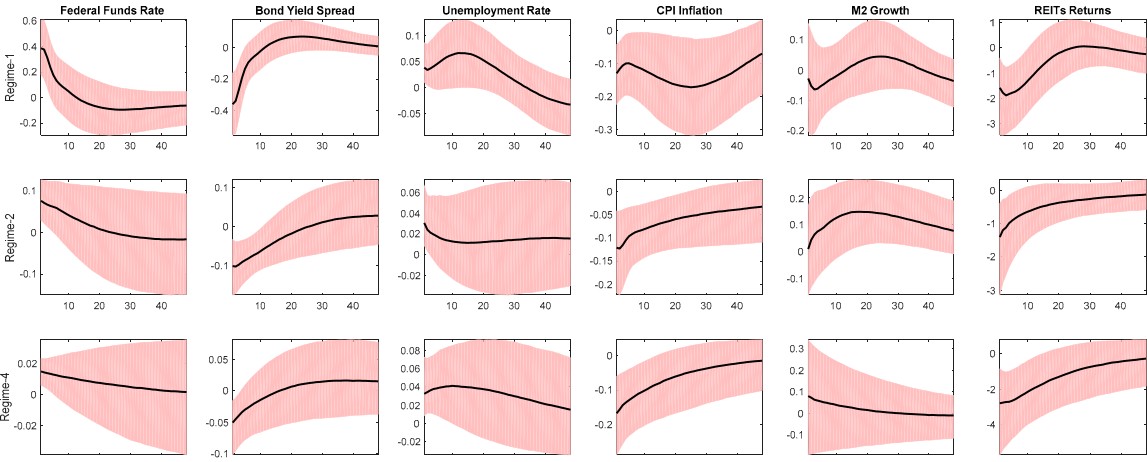

**Figure 3.** Impulse response functions to a contractionary monetary policy shock; Note: See Notes to Figure 1.

Next, Figure 4 presents the responses of the variables to a negative interest rate spread shock. To implement the analysis, we made it so that the short-term interest rate was exogenous to the spread shock in the third regime. Unsurprisingly, the positive impact on REITs returns tended to increase both in magnitude and length of the periods for which the effect was significant as we move from regimes 1 to 4. These results highlight the enhanced role of unconventional monetary policies in the third and fourth regimes aiming to reduce borrowing costs, as well as attempts made to directly stimulate the real estate sector, especially in the last identified sub-sample.

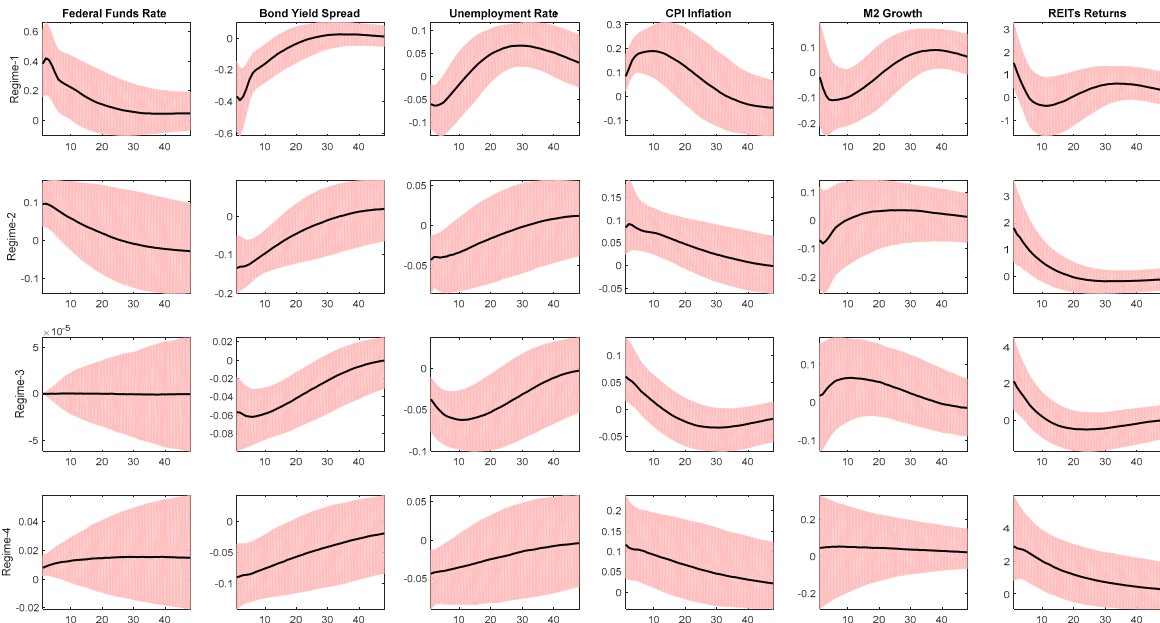

**Figure 4.** Impulse response functions to a negative interest rate spread shock; Note: See Notes to Figure 1.

Figure 5 presents the responses of the variables to an expansionary demand shock. The positive impact of REITs returns continued to increase in magnitude over the regimes, with the statistical significance also lasting for a longer number of horizons, especially in the last regime (where the effect was significant for over a year). While in Figure 6, we can see that the aggregate supply shock tended to positively affect the REITs returns, with the strongest effects observed in regimes 1, 3, and 4, with the effect declining during regime 2, with the effect being statistically longest-lasting (for over a year) in regime 3, i.e., right after the depth of financial crisis.

In summary, looking across all these impulse responses suggests that the transmission mechanism of the different shocks on REITs returns changed across the four regimes. To understand the extent to which movements of REITs returns can be explained by each shock and how the contribution of shocks changed across regimes, Figure 7 highlights the forecast error variance decompositions of the six endogenous variables for each of the four shocks. Concentrating on REITs returns, we observed that, the policy shock played an important role in the last regime (compared to the first two regimes), explaining over 30% of the variation in REITs returns. The spread shock was found to have quite a strong impact on regimes 2 and 4, with it explaining over 30% of the fluctuations in the real estate returns. Demand shocks explain about 14% and 8% of the variations in REITs returns in regimes 3 and 1, respectively. But the importance of the aggregate supply shocks stand out in regimes 1 and 2, with it explaining over 50% and 40% of the variations in regimes 1 and 2, and over 20% and 10% of the fluctuations of the REITs returns in regimes 3 and 4, respectively. The importance of the supply shocks is in line with Plakandaras et al. [11]. Overall, while supply shocks have been shown to be important in the early part of the sample, the spread and monetary shocks seem to dominate towards the end of the sample period under consideration.

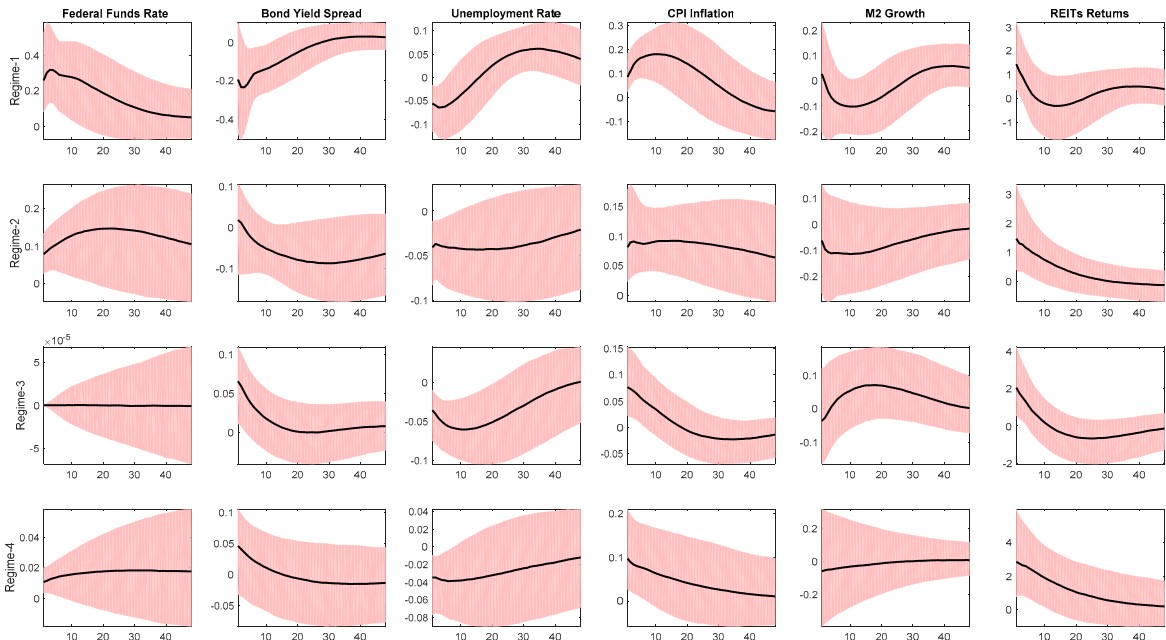

**Figure 5.** Impulse response functions to an expansionary demand shock; Note: See Notes to Figure 1.

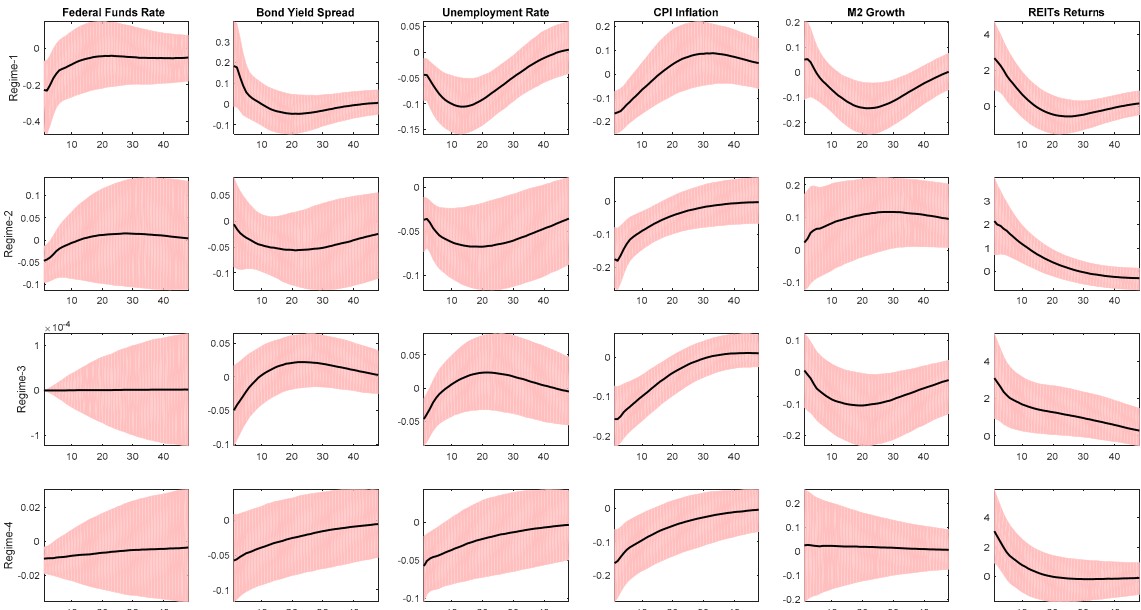

**Figure 6.** Impulse response functions to an expansionary supply shock; Notes: See Notes to Figure 1.

These results tend to suggest the importance of technical progress, i.e., the aggregate supply shocks, post-World War II in the US, which in turn led to a growth in productivity and hence output growth. This output growth is likely to have been driven by the consistently booming real estate market, until the collapse in 2007. The importance of the monetary policy and the spread shock in the last two regimes, basically coincided with the post-crisis period, where various measures of unconventional monetary policies were undertaken to boost the housing market and the overall macroeconomy. Our analysis thus indicates that while the role of monetary policy is important in driving the real estate sector of the US, it is also necessary to identify the role of other shocks, so that researchers do not overemphasize the role of monetary policy shocks.

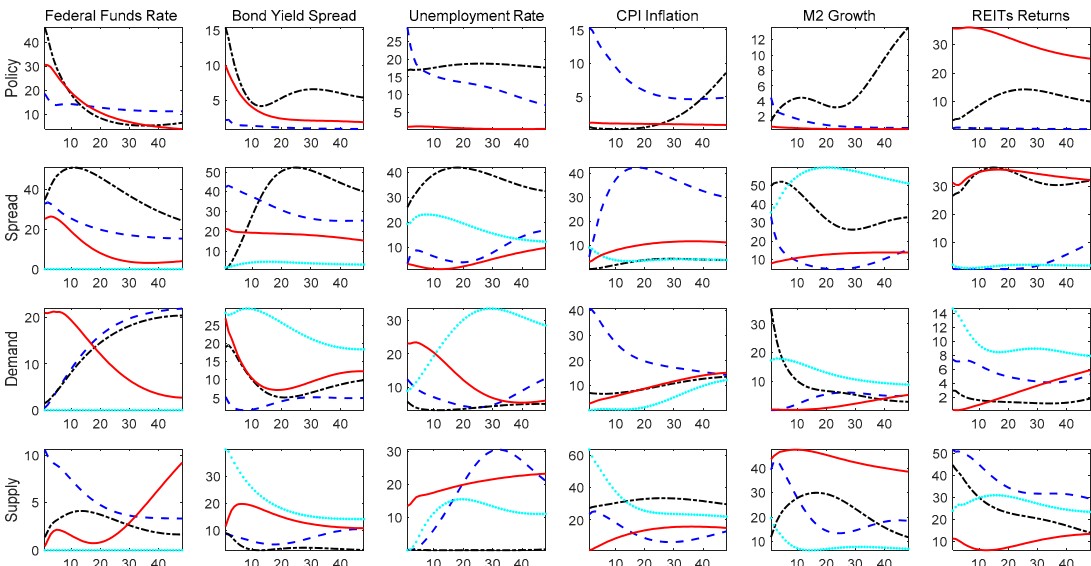

**Figure 7.** Forecast error variance decomposition; Notes: The four regimes correspond to the periods June 1973 to January 1985 (dashed blue line), February 1985 to January 2009 (dashed-dotted black line), February 2009 to May 2013 (dotted cyan line), and June 2013 to December 2016 (solid red line).

## 4. Conclusions

Given the importance of the real estate sector in causing the recent financial crisis, this paper uses a flexible change-point VAR model to analyze the regime-specific impact of various macroeconomic and financial shocks, identified based on sign-restrictions, on the US REITs returns. We deviate from the existing literature, which merely analyzes the role of monetary policy shocks on the US REITs sector and ignores the possible influence and importance of other shocks. The empirical model identifies three break points (four regimes) over the monthly sample period from 1972:12 to 2016:12. The third regime, coincides with the crisis period. The analysis discloses a range of important changes in the statistical and dynamic properties of REITs returns over the sample period. Statistical properties, such as persistence and volatility of fluctuations in REITs returns and the volatility of the reduced-form errors are found to have changed throughout the different regimes, with the crisis period being characterized by higher volatility. In addition, although quantitative changes are recorded throughout the whole period, supply, monetary policy, and spread shocks generate movements in REITs returns at the early and last parts of the sample period under consideration, respectively.

Hence, while the role of monetary policy is important in driving the real estate sector of the US, it is also necessary to identify the role of other shocks, in particular aggregate supply shocks to capture the role of productivity on the real estate sector, so that researchers do not overemphasize the role of monetary policy shocks. As we show, monetary policy shocks are only dominant in terms of moving the REITs sector post the recent financial crisis, in the wake of wide-array of unconventional monetary policy measures. This result also tends to suggest that, compared to productivity shocks, the role of monetary policy was minimal in heating up the US real estate market [30] before its collapse that led to the 'Great Recession'. Hence, loose monetary policy cannot be blamed for the real estate market bubble alone, though it is indeed true that post the financial liberalization in the US, the importance of monetary policy in affecting the real estate sector did increase [14].

From a policy perspective, our results have important implications. The relatively weaker role of conventional monetary policy, in affecting the REITs sector, tends to suggest that if there are bubbles in the US REITs sector, the Federal Reserve will not be successful in preventing it from bursting. In other words, besides the limited role of monetary policy on the REITs sector, the fact that interest rates are a blunt instrument to prick a bubble resulting in unintended collateral damage, the main policy message

from our analysis is that the Federal Reserve should focus on stabilizing inflation and the output gap only—an observation in line with Bernanke and Gertler [31,32].

One limitation of this study is that we consider the overall REITs sector of the US. However, REITs data is available in a sector-specific manner involving equities and mortgages, which in turn could be affected differently from these shocks compared to the overall market. These possible dissimilarities could be studied as part of future research. Moreover, we ignore the role of fiscal policies in the paper, which also played an important role in the US macroeconomy, especially during the ZLB. In addition, our current study can be extended by analyzing REITs markets of other developed countries or regions like the UK, Euro Area, and Japan, which also faced a ZLB situation. Finally, our approach can also be applied to study the possible heterogenous impact of these shocks across the regional housing markets of the US.

**Author Contributions:** Conceptualization, R.G.; Formal analysis, R.G., Z.L. and W.-K.W.; Methodology, R.G. and W.-K.W.; Software, Z.L.; Writing—original draft, R.G., Z.L. and W.-K.W.

**Funding:** This research received no external funding.

**Acknowledgments:** We would like to thank three anonymous referees for many helpful comments. However, any remaining errors are solely ours. The third author would like to thank Robert B. Miller and Howard E. Thompson for their continuous guidance and encouragement. This research has been supported by Northeast Normal University, The Education University of Hong Kong, Asia University, China Medical University Hospital, The Hang Seng University of Hong Kong, the Research Grants Council of Hong Kong (Project Number 12500915), and Ministry of Science and Technology (MOST, Project Numbers 106-2410-H-468-002 and 107-2410-H-468-002-MY3), Taiwan.

**Conflicts of Interest:** The authors declare no conflicts of interest.

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
