# Peer review of "Macroeconomic Shocks and Changing Dynamics of the U.S. REITs Sector"

_sustainability, doi:10.3390/su11102776_

Round 1

Reviewer 1 Report

The paper “Macroeconomic shocks and changing dynamics of the U.S. REITs sector” is a quite interesting paper. It investigates the role of various macroeconomic shocks (monetary policy shock, spread shock, demand shock and supply shock) on the prices in the Real Estate Investment Trusts prices in the USA. In the analysis a change-point vector autoregressive model is developed and four different regimes are identified in the period from 1973 to 2016.

In my opinion, several changes could be done in order to improve the quality of the paper and make it more interesting to readers.

1.     The goal of the paper should be clearly defined. What is the concrete question the study wants to answer? How is the study extending previous knowledge? In my opinion, these points should be explained in the abstract , the introduction and the conclusions section of the paper.

2.     Are there policy-making conclusions? Why is this investigation important? This should be clarified, specially in the conclusions section, which, in my opinion, should be extended. Many potential readers will first read the abstract and the conclusions of the paper, so it is not a bad idea to better explain the main findings in this section and devote more words to the explanation of the main findings and their importance. This is in line with point 1. The answer to the question the study wants to answer must be clearly stated in the conclusions section.

3.     Maybe the shortcomings of the research should be also mentioned in the conclusions section.

4.   Regarding citations. I believe the format employed in the text is not the correct one. As explained in the instructions for authors, references must be numbered in order of appearance in the text (including table captions and figure legends) and listed individually at the end of the manuscript. It is recommended to prepare the references with a bibliography software package, such as EndNote, ReferenceManager or Zotero to avoid typing mistakes and duplicated references. In the text, reference numbers should be placed in square brackets [ ], and placed before the punctuation; for example [1], [1–3] or [1,3].

Moreover, it is quite strange that 13 out of 30 references refer to the same author. Also the high number of “forthcoming documents”.

5.     I wonder if it is really necessary to include the foot notes and why this information cannot be included in the text

6.     Maybe it should be better explained what is the link of the paper and the concept of “sustainability”, unless this paper is for a special issue such as “Sustainability Challenges in Real Estate Markets and Urban Property Developments”.

In general, the topic of the paper is interesting, but the in my opinion the changes proposed are important to enhance the presentation of the research.

Author Response

Response to Reviewer 1 Comments

The paper “Macroeconomic shocks and changing dynamics of the U.S. REITs sector” is a quite interesting paper. It investigates the role of various macroeconomic shocks (monetary policy shock, spread shock, demand shock and supply shock) on the prices in the Real Estate Investment Trusts prices in the USA. In the analysis a change-point vector autoregressive model is developed and four different regimes are identified in the period from 1973 to 2016.

In my opinion, several changes could be done in order to improve the quality of the paper and make it more interesting to readers. 

1.     The goal of the paper should be clearly defined. What is the concrete question the study wants to answer? How is the study extending previous knowledge? In my opinion, these points should be explained in the abstract, the introduction and the conclusions section of the paper.

Authors’ Response: We thank the referee for this comment. This issue has now been addressed in the abstract, introduction (last paragraph of this section in particular) and conclusion.

2.     Are there policy-making conclusions? Why is this investigation important? This should be clarified, specially in the conclusions section, which, in my opinion, should be extended. Many potential readers will first read the abstract and the conclusions of the paper, so it is not a bad idea to better explain the main findings in this section and devote more words to the explanation of the main findings and their importance. This is in line with point 1. The answer to the question the study wants to answer must be clearly stated in the conclusions section.

Authors’ Response: We thank the referee for this comment. This issue has now been addressed. Please refer to the second and third paragraph in the conclusion segment.

3.     Maybe the shortcomings of the research should be also mentioned in the conclusions section.

Authors’ Response: We thank the referee for this comment. We have now discussed the limitations of our research in the last paragraph of the conclusion segment.

4.   Regarding citations. I believe the format employed in the text is not the correct one. As explained in the instructions for authors, references must be numbered in order of appearance in the text (including table captions and figure legends) and listed individually at the end of the manuscript. It is recommended to prepare the references with a bibliography software package, such as EndNote, ReferenceManager or Zotero to avoid typing mistakes and duplicated references. In the text, reference numbers should be placed in square brackets [ ], and placed before the punctuation; for example [1], [1–3] or [1,3].

Moreover, it is quite strange that 13 out of 30 references refer to the same author. Also the high number of “forthcoming documents”.

Authors’ Response: We thank the referee for this comment. This issue has now been addressed. Regarding the fact that many references are from the same author is basically because one of the authors in this paper has indeed done a lot of work in the area related to the US real estate sector and these papers also contain the most recent literature reviews, which the readers can access. Self-citation is not the objective here, as the papers that have been cited are indeed important and relevant to the topic being analysed in this paper. We have now, wherever possible, updated the information on the forthcoming papers with appropriate references or DOIs to indicate that indeed the forthcoming papers are already published or going to be published in the future respectively.

5.     I wonder if it is really necessary to include the foot notes and why this information cannot be included in the text

Authors’ Response: We thank the referee for these comments. The idea behind the footnotes is to include important information related to the broader topic (in this case the REITs market) but not necessarily related to the paper. We have, however, incorporated one of the footnotes into the text.

6.     Maybe it should be better explained what is the link of the paper and the concept of “sustainability”, unless this paper is for a special issue such as “Sustainability Challenges in Real Estate Markets and Urban Property Developments”.

Authors’ Response: We thank the referee for this comment, but we are not sure as to where this information can be directly incorporated into the paper. But recall, the Special Issue to which the paper was submitted to is titled as: “Sustainability of the Theories Developed by Mathematical Finance and Mathematical Economics with Applications”. In other words, our paper fits into the category of Applications to ensure the Sustainability of the Theories developed by Mathematical Economics, which in our case is the change-point VAR. In addition, since we are dealing with the REITs sectors, our paper is also associated with Mathematical Finance.

 In general, the topic of the paper is interesting, but in my opinion, the changes proposed are important to enhance the presentation of the research.

Authors’ Response: We thank the referee for her/his valuable comments and we hope that we have addressed all of her/his concerns in the revised version of the paper.

Reviewer 2 Report

The paper examines how various macroeconomic and financial shocks explain Real Estate Investment Trusts prices. The authors find that the transmission mechanisms of the shocks vary across different regimes. The authors need to address the following points before the paper is considered for publication:

1.       They need to describe in the paper how they extract the structural shocks: monetary shocks, demand shocks, and the other shocks. At least, provide intuition on the identification of these shocks.

2.       I do not see Regime 3 in Figure 3.

3.       They should add the implications of the findings in the conclusion section.

4.       It seems the response of REIT prices to monetary shocks was stronger only in Regime 4. Based on this, can we say monetary policy did not play a role in driving up the housing prices prior to the Great Recession? Are the findings suggesting that the other shocks mattered more?

Author Response

Response to Reviewer 2 Comments

The paper examines how various macroeconomic and financial shocks explain Real Estate Investment Trusts prices. The authors find that the transmission mechanisms of the shocks vary across different regimes. The authors need to address the following points before the paper is considered for publication:

1.         They need to describe in the paper how they extract the structural shocks: monetary shocks, demand shocks, and the other shocks. At least, provide intuition on the identification of these shocks.

Authors’ Response: We thank the referee for this comment. This has now been addressed. Please refer to the paragraph immediately below Table 1.

2.         I do not see Regime 3 in Figure 3.

Authors’ Response: We thank the referee for this comment. Note that in Regime 3, with it coinciding with the zero lower bound, the monetary policy variable was set at zero, and hence, there is no monetary policy shock in Regime 3. We have now made this explicit in the paper. Please refer to the second paragraph below Figure 2.

3.         They should add the implications of the findings in the conclusion section.

Authors’ Response: We thank the referee for this comment. This has now been added in the second paragraph of the conclusion segment.

4.         It seems the response of REIT prices to monetary shocks was stronger only in Regime 4. Based on this, can we say monetary policy did not play a role in driving up the housing prices prior to the Great Recession? Are the findings suggesting that the other shocks mattered more?

Authors’ Response: We thank the referee for this comment. A discussion in this regard has now been added to the second paragraph of the conclusion segment.

Reviewer 3 Report

Summary

This paper uses an innovative VAR model to investigate how a variety of shocks affect REITs returns in the United States. In addition, the authors, using monthly data, conduct a regime analysis that identifies four separate macroeconomic regimes in the United States since the start of the sample period (1972). The authors identify a few key results. First, increases in the federal funds rate caused REITs returns to go down. Decreases in the term spread (difference between 10-year Treasury rate and federal funds rate) increase REIT returns. Lastly, positive shocks to aggregate demand and aggregate supply increase REITs returns.

Positive Aspects

This is a solid paper using an innovative VAR model. I have only minor suggestions.

Areas for Improvement

1) I always recommend that authors do a thorough read through for minor grammatical mistakes.

2) One thing that I think is a little lacking is an explanation of the main results (Figure 7 in particular). The authors briefly describe the differences across the regimes in terms of the REITs returns. However, they don’t offer much explanation of why the regimes are different. This doesn’t have to be a theoretical explanation just a description of why the authors think the regimes differ. I would say a paragraph or so would be sufficient (and any citations that the authors deem relevant).

Author Response

Response to Reviewer 3 Comments

Summary

This paper uses an innovative VAR model to investigate how a variety of shocks affect REITs returns in the United States. In addition, the authors, using monthly data, conduct a regime analysis that identifies four separate macroeconomic regimes in the United States since the start of the sample period (1972). The authors identify a few key results. First, increases in the federal funds rate caused REITs returns to go down. Decreases in the term spread (difference between 10-year Treasury rate and federal funds rate) increase REIT returns. Lastly, positive shocks to aggregate demand and aggregate supply increase REITs returns.

Positive Aspects

This is a solid paper using an innovative VAR model. I have only minor suggestions.

Authors’ Response: We thank the referee for liking our paper.

Areas for Improvement

1) I always recommend that authors do a thorough read through for minor grammatical mistakes.

Authors’ Response: We thank the referee for this comment. We have gone through the paper and have checked for possible grammatical errors.

2) One thing that I think is a little lacking is an explanation of the main results (Figure 7 in particular). The authors briefly describe the differences across the regimes in terms of the REITs returns. However, they don’t offer much explanation of why the regimes are different. This doesn’t have to be a theoretical explanation just a description of why the authors think the regimes differ. I would say a paragraph or so would be sufficient (and any citations that the authors deem relevant).

Authors’ Response: We thank the referee for this comment. Please refer to the paragraph before Figure 7, where we have now provided an explanation regarding the importance of the various shocks across the regimes identified.

Round 2

Reviewer 1 Report

I would like to congratulate the authors.

They have done a good job. I really believe that the paper is much clearer now. The objectives of the paper are directly described in the abstract, as well as the main findings. Also the originality of the paper and the policy-making conclusions are more obvious for readers. 

In fact, one of the main findings, which is the interest and need of monitoring other shocks other than monetary shocks, like demand and supply shocks has been appropriately underlined in this new version of the paper.